# One Health—Its Importance in Helping to Better Control Antimicrobial Resistance

**DOI:** 10.3390/tropicalmed4010022

**Published:** 2019-01-29

**Authors:** Peter J. Collignon, Scott A. McEwen

**Affiliations:** 1Infectious Diseases and Microbiology, Canberra Hospital, Garran, ACT 2605, Australia; 2Medical School, Australian National University, Acton ACT 2601, Australia; 3Department of Population Medicine, University of Guelph, Guelph N1G 2W1, Canada; smcewen@uoguelph.ca

**Keywords:** One Health, antibiotics, antimicrobials, antimicrobial resistance, environment, water, infrastructure

## Abstract

Approaching any issue from a One Health perspective necessitates looking at the interactions of people, domestic animals, wildlife, plants, and our environment. For antimicrobial resistance this includes antimicrobial use (and abuse) in the human, animal and environmental sectors. More importantly, the spread of resistant bacteria and resistance determinants within and between these sectors and globally must be addressed. Better managing this problem includes taking steps to preserve the continued effectiveness of existing antimicrobials such as trying to eliminate their inappropriate use, particularly where they are used in high volumes. Examples are the mass medication of animals with critically important antimicrobials for humans, such as third generation cephalosporins and fluoroquinolones, and the long term, in-feed use of antimicrobials, such colistin, tetracyclines and macrolides, for growth promotion. In people it is essential to better prevent infections, reduce over-prescribing and over-use of antimicrobials and stop resistant bacteria from spreading by improving hygiene and infection control, drinking water and sanitation. Pollution from inadequate treatment of industrial, residential and farm waste is expanding the resistome in the environment. Numerous countries and several international agencies have now included a One Health Approach within their action plans to address antimicrobial resistance. Necessary actions include improvements in antimicrobial use, better regulation and policy, as well as improved surveillance, stewardship, infection control, sanitation, animal husbandry, and finding alternatives to antimicrobials.

## 1. Introduction

Antimicrobial resistance is a global public health problem [1,2]. Most bacteria that cause serious infections and could once be successfully treated with several different antibiotic classes, have now acquired resistance—often to many antibiotics. In some regions the increased resistance has been so extensive that resistance is present in some bacteria to nearly all of these drugs [2,3,4]. The threat is most acute for antibacterial antimicrobials (antibiotics—the focus of this paper) but also threatens antifungals, antiparastics and antivirals [5]. 

Antimicrobial overuse is occurring in multiple sectors (human, animal, agriculture) [3,6]. Microorganisms faced with antimicrobial selection pressure enhance their fitness by acquiring and expressing resistance genes, then sharing them with other bacteria and by other mechanisms, for example gene overexpression and silencing, phase variation. When bacteria are resistant they also present in much larger numbers when exposed to antimicrobials, whether in an individual, in a location and in the environment. Additionally important in driving the deteriorating resistance problem are factors that promote the spread of resistant bacteria (or “contagion”) [7]. This spread involves not only bacteria themselves but the resistance genes they carry and that can be acquired by other bacteria [8]. Factors that facilitate “contagion” include poverty, poor housing, poor infection control, poor water supplies, poor sanitation, run off of waste from intensive agriculture, environmental contamination and geographical movement of infected humans and animals [9,10,11]. 

Wherever antimicrobials are used, there are often already large reservoirs of resistant bacteria and resistance genes. These include people and their local environments (both in hospitals and in the community), as well as animals, farms and aquaculture environments. Large reservoirs of resistance and residual antimicrobials occur in water, soil, wildlife and many other ecological niches, not only due to pollution by sewage, pharmaceutical industry waste and manure runoff from farms [10,12,13], but often resistant bacteria and resistance genes have already been there for millennia [14,15].

Most bacteria and their genes can move relatively easily within and between humans, animals and the environment. Microbial adaptations to antimicrobial use and other selection pressures within any one sector are reflected in other sectors [8,16]. Similarly, actions (or inactions) to contain antimicrobial resistance in one sector affect other sectors [17,18]. Antimicrobial resistance is an ecological problem that is characterized by complex interactions involving diverse microbial populations affecting the health of humans, animals and the environment. It makes sense to address the resistance problem by taking this complexity and ecological nature into account using a coordinated, multi-sectoral approach, such as One Health [5,19,20,21,22,23]. 

One Health is defined by WHO [24] and others [25] as a concept and approach to “designing and implementing programs, policies, legislation and research in which multiple sectors communicate and work together to achieve better public health outcomes. The areas of work in which a One Health approach is particularly relevant include food safety, the control of zoonoses and combatting antibiotic resistance” [24]. It needs to involve the “collaborative effort of multiple health science professions, together with their related disciplines and institutions—working locally, nationally, and globally—to attain optimal health for people, domestic animals, wildlife, plants, and our environment” [25]. The origins of One Health are centuries old and are based on the mutual inter-dependence of people and animals and a recognition that they share not only the same environment, but also many infectious diseases [23]. Our current concept of One Health however goes much further. It also embraces the health of the environment. 

## 2. Use of Antimicrobials in Humans, Animals and Plants

The vast majority of antimicrobial classes are used both in humans and animals (including aquaculture; both farmed fish and shellfish). Only few antimicrobial classes are reserved exclusively for humans (e.g., carbapenems). There are also few classes limited to veterinary use (e.g., flavophospholipols, ionophores); mainly because of toxicity to humans [26,27,28,29,30].

Insects (e.g., bees) and some plants are frequently treated with antimicrobials. Tetracyclines, streptomycin and some other antimicrobials are used for treatment and prophylaxis of bacterial infections of fruit, such as apples and pears (e.g., “fire blight” caused by *Erwinia amylovora*) [31,32]. Antifungals, especially azoles, are used in huge quantities and applied to broad acre crops such as wheat [33]. 

There are marked differences in the ways antimicrobials are used in human compared to non-human sectors. In people, antimicrobials are mostly used for treatment of clinical infections in individual patients, with some limited prophylactic use in individuals (e.g., post-surgery) or occasionally in groups (e.g., prevention of meningococcal disease). Antimicrobial uses in companion animals (e.g., dogs, cats, pet birds, horses) are broadly similar to those in humans, with antimicrobials mostly administered on an individual basis to treat infection, and occasionally for prophylaxis, such as post-surgery [34,35]. 

In the food-producing animal sector, antimicrobials are also used therapeutically to treat individual clinically sick animals (e.g., dairy cows with mastitis) [26]. However, in intensive farming and aquaculture, for reasons of practicality and efficiency, antimicrobials are often administered through feed or water to entire groups (e.g., pens of pigs, flocks of broilers), either for prophylaxis (to healthy animals at risk of infection) or metaphylaxis (to healthy animals in the same group as diseased animals) [36]. Some have even succeeded in having this group level administration defined (and we believe inappropriately) in the animal health sector as “therapeutic” use. Growth promotion, prophylaxis and metaphylaxis account for by far the largest volumes of antimicrobials used in the food-producing animal sector [26,27,37]. 

### Growth Promotion Use

Using antimicrobials for growth promotion is highly controversial because instead of treating sick animals they are administered to healthy animals, usually for prolonged periods of time, and often at sub-therapeutic doses in order to improve production. These conditions favor selection and spread of resistant bacteria within animals and to humans through food or other environmental pathways [38,39]. The period of exposure with growth promotion is usually greater than two weeks and often almost the entire life of an animal, for example in chicken for 36 days or more. 

Based on studies, mostly conducted decades ago, the purported production benefits of antimicrobial growth promoters range widely (1–10%). Surveillance and animal production data however now suggests that benefits in animals reared in good conditions are probably quite small and may be non-existent. Many large poultry corporations are now marketing chicken raised without antimicrobials administered at hatchery or farm levels [40]. Expressed concerns are that antimicrobial growth promoters are used to compensate for poor hygiene and housing, and as replacement for proper animal health management [18,41,42]. For these reasons, the World Health Organization (WHO) advocates the termination of antimicrobial use for growth promotion [5,41]. This practice has now been banned in Europe and elsewhere and is being phased out in some other countries [43,44,45]. However there are still many countries where they continue to be used [46], including drugs categorized by WHO as critically important to humans, for example colistin, fluoroquinolones and macrolides [47].

Comprehensive global quantitative data on use of antimicrobial agents in humans, animals and plants is generally lacking. Table 1 shows the varying levels of antibiotic usage in people around the world, associated resistance levels, plus some social and infrastructure parameters—the latter of which can facilitate the spread of resistant bacteria (e.g., poor sanitation). Figure 1 shows antibiotic use in different regions globally in people and the lack of correlation with increased resistance levels in bacteria and human antibiotic usage. These data strongly suggest that there are other very important factors influencing antimicrobial resistance over and above simply antibiotic usage.

Aggregating countries into regional groupings shows a pattern where there is an inverse aggregate relationship between antimicrobial resistance and usage. These data help confirm that there are other very important factors influencing antimicrobial resistance over and above simply antibiotic usage. (Figures assembled from data taken from reference 7)

The World Organization for Animal Health has developed a global database on the use of antimicrobial agents in animals [46]. Figure 2 shows reported quantities of antimicrobials used in animals in 2014, summarized by OIE Region and expressed as total quantities (tons) and adjusted for animal biomass. Additionally, included is the per cent of countries authorizing the use of antimicrobials for growth promotion. Tetracyclines accounted for the largest proportion of overall antimicrobial use globally (37.1% of total), followed by polypeptides (15.7%), penicillins (9.8%), macrolides (8.9%) and aminoglycosides (7.8%) [46]. 

## 3. One Health Antimicrobial Resistance Case Studies

The following examples illustrate antimicrobial resistance problems that arise when the same classes of antimicrobials are used in humans and animals, and the challenges that arise from competing interests and imbalances of risk and benefit in various sectors. 

### 3.1. Third Generation Cephalosporins

Third generation cephalosporins are broad spectrum beta-lactam antimicrobials that are widely used in humans and animals. In people, cefotaxime, ceftriaxone and several other members of the class are used for a wide variety of frequently serious infections, particularly in hospital settings, for example bloodstream infections due to *Escherichia coli* and other bacteria, but also in community settings, for example *Neisseria gonorrhea* [47]. Third generation cephalosporins are classified as “critically important” for human health [47]. 

Ceftiofur is the principal third generation cephalosporin for veterinary use; others include cefpodoxime, cefoperazone and cefovecin. Ceftiofur is injected and used in animals as therapy to treat pneumonia, arthritis, septicemia and other conditions [48,49]. However ceftiofur is also used in mass therapy (metaphylaxis or prophylaxis), either under an approved label claim (e.g., injection of feedlot cattle for control of bovine respiratory disease), or off-label (e.g., injection of hatching eggs or day-old chicks for prevention of *E. coli* infections). Factors that encourage overuse of ceftiofur are its broad spectrum activity, zero withdrawal time for milk from dairy animals (due to its high maximum residual level; MRL), and availability of a long-acting preparation [48,49]. 

In Europe, approximately 14 tons of third and fourth generation cephalosporins were used in 2014 for use in animals [28]. Similar volumes are used in the US [50]. In many countries, cephalosporins are commonly used in humans but with wide variations. Overall, 101 tons of third generation cephalosporins were used in people Europe in 2012 [29] and in the US, approximately 82 tons in 2011 [51]. 

Resistance to the third generation cephalosporins is mainly mediated by extended-spectrum beta-lactamases (ESBLs) and AmpC beta-lactamases [47]. ESBL genes are highly mobile and transmitted on plasmids, transposons and other genetic elements. AmpC beta-lactamases were originally reported to be chromosomal but have also been identified on plasmids and to have spread horizontally among *Enterobacteriaceae* [47]. Unfortunately, in many countries resistance to third generation cephalosporins is now common among *E. coli* and *K. pneumonia* [52,53]. Resistance genes are frequently co-located with genes encoding resistance to other classes of antimicrobials, including tetracyclines, aminoglycosides and sulfonamides. As a consequence, the use of other antimicrobials in animals, for example tetracyclines administered in feed, can select for ESBL strains of bacteria [54]. 

Ceftiofur can be administered to eggs or day-old chicks in hatcheries, using automated equipment that injects small quantities of the drug into the many thousands of hatching eggs or chicks intended for treated flocks as prophylaxis against *E. coli* infections [55,56]. This practice selected for cephalosporin resistance in *Salmonella* Heidelberg, an important cause of human illness and associated with consumption of poultry products [57]. Surveillance detected a high degree of time-related correlations in trends of resistance to ceftiofur (and ceftriaxone, a drug of choice for treatment of severe cases of salmonellosis in children and pregnant women) among *Salmonella* Heidelberg from clinical infections in humans, from poultry samples collected at retail, and in *E. coli* from retail poultry samples [55]. Voluntary termination of ceftiofur use in hatcheries in Quebec was followed by a precipitous drop in the prevalence of resistance to ceftiofur. Subsequent re-introduction of its use, was followed by a return to higher prevalence of resistance [56]. In recognition of the resultant human health risks, in 2014, the Canadian poultry industry placed a voluntary ban on the use of ceftiofur and other critically important antimicrobials for disease prophylaxis [58]. 

In Japan, voluntary withdrawal of the off-label use of ceftiofur in hatcheries in 2012 was also followed by a significant decrease in broad-spectrum cephalosporin resistance in *E. coli* from broilers [59]. Some other countries (e.g., Denmark and Australia) have also placed voluntary restrictions in its use [60]. The label claim for day-old injection of poultry flocks was withdrawn in Europe, while some countries banned off-label use of third generation cephalosporins (e.g., U.S.) [48,61], and in other countries there is a requirement that use is restricted to situations where no other effective approved drugs are available for treatment [62]. 

### 3.2. Colistin

Colistin is in the polymyxin class of antimicrobials, and has been used in both people and animals for over 50 years [63]. Polymyxins, when administered systemically, frequently cause nephrotoxicity and neurotoxicity in people [64]. Thus, until recently its use was mainly limited to topical use and the treatment of infections in cystic fibrosis patients by inhalation (with a colistimethate sodium). 

Colistin however is now used much more frequently, as a drug of last resort by injection, for treatment of multi-resistant gram-negative infections including carbapenem-resistant *Pseudomonas aeruginosa* and *E. coli* [65,66,67]. Where approved for use in food animals (e.g., Brazil, Europe, China), most colistin is administered orally to groups of pigs, poultry and in some cases calves, for treatment and prophylaxis of diarrhea due to gram-negative infections or for growth promotion [63,67,68]. In countries where data are available, the quantities consumed for animal production vastly exceed those used in humans and is very variable between countries [69]. In 2013 total animal consumption in Europe was 495 tons; 99.7% in oral form (e.g., for oral solution, medicated feed premix and oral powder) [63]. In China, the world’s largest producer of pigs and poultry, an estimated 12,000 tons of colistin was used in food animals [68]. 

Until recently, limited data on colistin resistance were available, partly because of technical difficulties in phenotypic susceptibility testing [63,70]. In Europe in 2016, resistance was found in 1.9% of indicator *E. coli* from broilers, 3.9% from broiler meat, 6.1% from turkeys and 10.1% from turkey meat [71]. Colistin resistance was thought limited to chromosomal mutation and was essentially non-transferable [63], however in 2015 the transferable plasmid-mediated colistin resistance gene, mcr-1, was found in *E. coli* isolates obtained from animals, food and human bloodstream infections from China [68]. Spread of the gene by conjugation has been shown in *Klebsiella pneumoniae*, *Enterobacter aerogenes*, *Enterobacter spp*. and *P. aeruginosa* [68]. Retrospective analyses have demonstrated the mcr-1 gene in several bacterial species isolated from humans, animals and environmental samples in numerous countries [72,73,74,75,76], and the gene was found in about 5% of healthy travelers [77]. The earliest identification of the gene thus far was in *E. coli* from poultry collected in the 1980s in China [78]. The mcr-1 gene has also been detected in isolates obtained from wildlife and surface water samples, demonstrating environmental contamination [79]. Recently, other plasmid-mediated colistin resistance genes has been reported for example mcr-2 in *E. coli* from pigs in Belgium [80]. 

Colistin illustrates some important One Health dimensions of antimicrobial resistance that differ from those of third generation cephalosporins. The toxicity with systemic use and the availability of other safer and more effective antimicrobials, meant for many years colistin was mainly used topically in people. However with the emergence of multi-drug resistance in many Gram-negative bacteria, there has been increasing need for this drug to systemically treat severe, life-threatening infections in humans in many countries. The colistin case demonstrates (once again) that using large quantities antimicrobials for group treatments or growth promotion in animals can lead to significant antimicrobial resistance problems for human health, even if the drug class is initially believed to be of lesser importance, because the relative importance of antimicrobials to human health can change. This is the same problem that arose from using avoparcin as a growth promoter until it was banned; it selected for resistance to another glycopeptide, vancomycin, which is used for the treatment of life-threatening MRSA (methicillin resistant *Staphylococcus aureus*) and for treating serious enterococcal infections (the latter especially in penicillin allergic patients) [81,82]. 

## 4. Risks to Public Health and Animal Health

Antimicrobial resistance is harmful to health because it reduces the effectiveness of antimicrobial therapy and tends to increase the severity, incidence and costs of infection [3,83]. There is now considerable evidence that antimicrobial use in animals is an important contributor to antimicrobial resistance among some pathogens of humans, in particular, common enteric pathogens such as *Salmonella spp.*, *Campylobacter spp.*, *Enterococcus spp.* and *E. coli* [6,18,26,38,41]. 

Non-typhoidal *Salmonella* (NTS) are among the most common bacteria isolated from foodborne infections of humans. Globally, there are approximately 94 million cases, including 155,000 deaths each year [1]. Animals are the most important reservoirs of NTS for humans [38,84,85,86]. Fecal shedding by carrier animals is an important source of antimicrobial resistant *Salmonella* contamination of meat and poultry products [38], and may also be responsible for fruit and vegetable contamination through fecal contamination of the environment [87]. *Salmonella* resistance to any medically important antimicrobial is of concern, but particularly to those critically important to human health, such as cephalosporins and fluoroquinolones [38,41,56]. Therapy in some groups (e.g., children and pregnant women) can be very restricted and beta-lactams such as third generation cephalosporins often may be the only therapy available to treat serious infections. 

From the One Health antimicrobial resistance perspective, the third generation cephalosporins are good examples of antimicrobials that are considered critically important for both human and animal health. The main concern regarding selection and spread of resistance from animals to humans is their use as mass medications in large numbers of animals, either for therapy or prophylaxis. There are parallels with fluoroquinolones, another class of critically important antimicrobials, to which resistance among *Campylobacter jejuni* emerged following mass medication of poultry flocks [88,89,90]. In Australia where fluoroquinolones were never approved in food animals, fluoroquinolone resistant strains in food animals remain very rare [91]. 

Fluoroquinolone use in food animals is also linked to quinolone resistance in *Salmonella* [41,92,93,94]. Surveillance data compiled by WHO indicate that rates of fluoroquinolone resistance in non-typhoidal *Salmonella* vary widely by geographical region. For example, rates are relatively low in Europe (2–3%), higher in the Eastern Mediterranean region (up to 40–50%) and wide ranging in the Americas (0–96%) (1). Many *Salmonella* are also resistant to antimicrobials that have long been used as growth promoters in many countries (e.g., Canada, USA) including tetracyclines, penicillins and sulfonamides [41,84]. Antimicrobial resistance in some of the more virulent *Salmonella* serovars (e.g., Heidelberg, Newport, Typhimurium) has been associated with more severe infections in humans [38,83,86,95]. Resistance to other critically important antimicrobials continues to emerge in *Salmonella*, for example, a carbapenem resistant strain of *Salmonella* was identified on a pig farm that routinely administered prophylactic cephalosporin (ceftiofur) to piglets [96].

*Escherichia coli* are important pathogens of both humans and animals. In humans, *E. coli* are a common cause of serious bacterial infections, including enteritis, urinary tract infection and bloodstream infections [97,98,99]. Currently in England the rate for blood stream infections is about 64 cases per 100,000 per year and rising. A large and increasing proportion involves antimicrobial resistance, including fluoroquinolone resistance [100]. These higher rates are also being seen in countries with good surveillance systems in place, for example Denmark [60]. 

Many *E. coli* appear to behave as commensals of the gut of animals and humans, but may be opportunistic pathogens as well as donors of resistance genetic elements for pathogenic *E. coli* or other species of bacteria [101,102]. Although antimicrobial resistance is a rapidly increasing problem in *E. coli* infections of both animals and humans, the problem is better documented for isolates from human infections, where resistance is extensive, particularly in developing countries [1,103]. Humans are regularly exposed to antimicrobial resistant *E. coli* through foods and inadequately treated drinking water [104,105]. 

Travelers from developed countries are at risk of acquiring multi-resistance *E. coli* from other people or contaminated food and/or water [97,105,106]. There are now serious problems with extended spectrum beta lactamase (ESBL) *E. coli* in both developing and developed countries and foods from animals, in particular poultry, have been implicated as sources for humans [99,107,108], although the magnitude of the contribution from food animals is uncertain [102,103,104]. 

Given the critical importance of third and fourth generation cephalosporins and fluoroquinolones to human medicine and the clear evidence that treatment of entire groups of animals selects for resistance in important pathogens that spread from animals to humans [56,90], these drugs should be used rarely, if at all in animals, and only when supporting laboratory data demonstrate that no suitable alternatives of lesser human health importance are available. Their use as mass medications should be restricted.

Serious staphylococcal infections in people are common, including with Methicillin-resistant *Staphylococcus aureus* (MRSA) in both community and hospital settings, causing skin, wound, bloodstream and other types of infection [1,109,110,111]. *Staphylococcus aureus* and other staphylococci are also recognized pathogens of animals, for example they are responsible for cases of mastitis in cattle, and skin infections in pigs and companion animals [112,113]. MRSA were until recently relatively rare in animals but strains pathogenic to humans have emerged in several animal species [113,114,115,116]. Transmission to humans is thought currently to be mainly through contact with carrier animals [116]. The predominant strain isolated from animals is sequence type (ST) 398, and while pathogenic to humans, it is not considered a major epidemic strain [112,113]. Antimicrobial use in livestock, as well as lapses in biosecurity within and between farms, and international trade in animals, food or other products, are factors contributing to the spread of this pathogen in animals [113,117]. 

## 5. One Health Considerations from the Environment

One Health includes consideration of the environment as well as human and animal health [23,111]. The ecological nature of antimicrobial resistance is a reflection and consequence of the interconnectedness and diversity of life on the planet [22]. Many pathogenic bacteria, the antimicrobials that we use to treat them, and genes that confer resistance, have environmental origins (e.g., soil) [8,14,20]. Some important resistance genes, such as beta lactamases, are millions of years old [14,15]. Soil and other environmental matrices are rich sources of highly diverse populations of bacteria and their genes [14,118]. Antimicrobial resistance to a wide variety of drugs has been demonstrated in environmental bacteria isolated from the pre-antibiotic era, as well as from various sites (e.g., caves) free of other sources of exposure to modern antimicrobials [8,15,111,119]. Despite having ancient origins, there is abundant evidence that human activity has an impact on the resistome, which is the totality of resistance genes in the wider environment [13,14,118,119]. Hundreds of thousands of tons of antimicrobials are produced annually and find their way into the environment [18,27]. Waste from treatment plants and pharmaceutical industry, particularly if inadequately treated, can release high concentrations of antimicrobials into surface water [18,19,120,121]. Residues of antimicrobials are constituents of human sewage, livestock manure, and aquaculture, along with fecal bacteria and resistance genes [118,122,123,124,125]. Sewage treatment and composting of manure reduce concentrations of some but not all antimicrobials and microorganisms, which are introduced to soil upon land application of human and animal bio-solids [126]. 

Various environmental pathways are important routes of human exposure to resistant bacteria and their genes from animal and plant reservoirs [18,96,127] and provide opportunities for better regulations to control antimicrobial resistance. In developed countries with good quality sewage and drinking water treatment, and where most people have little to no direct contact with food-producing animals, transmission of bacteria and resistance genes from agricultural sources is largely foodborne, either from direct contamination of meat and poultry during slaughter and processing, or indirectly from fruit and vegetables contaminated by manure or irrigation water [38,87,90]. 

In countries with poor sewage and water treatment, drinking water is likely to be very important in transmission of resistant bacteria and/or genes from animals [11,97,111,120]). Poor sanitation also facilitates indirect person–person waterborne transmission of enteric bacteria among residents as well as international travelers who then return home colonized with resistant bacteria acquired locally [103,128]. Through these and other means, including globalized trade in animals and food, and long-distance migratory patterns of wildlife, antimicrobial resistant bacteria are globally disseminated.

General measures to address antimicrobial resistance in the wider environment include improved controls on pollution from industrial, residential and agricultural sources. Improved research as well as environmental monitoring and risk assessment is required to better understand the role of the environment in selection and spread of antimicrobial resistance, and to identify more specific measures to address resistance in this sector [12,14,18,100,103,129].

## 6. One Health Strategies to Address Antimicrobial Resistance

WHO and other international agencies (e.g., Food and Agriculture Organization (FAO), World Organization for Animal Health (OIE)), along with many individual countries, have developed comprehensive action plans to address the antimicrobial resistance crisis [5,130,131,132,133,134,135,136]. The WHO Global Action Plan seeks to address five major objectives that comprise the subtitles of the following sections. The WHO Plan embraces a One Health approach to address antimicrobial resistance, and it calls on member countries to do the same when developing their own action plans (6). There are five main pillars to the WHO Global Plan:Improve Awareness and Understanding of Antimicrobial Resistance through Effective Communication, Education and TrainingStrengthen the Knowledge and Evidence Base through Surveillance and ResearchReduce the Incidence of Infection through Effective Sanitation, Hygiene and Infection Prevention MeasuresOptimize the Use of Antimicrobial Medicines in Human and Animal HealthDevelop the Economic Case tor Sustainable Investment that Takes Account of the Needs of All Countries, and Increase Investment in New Medicines, Diagnostic Tools, Vaccines and Other Interventions

The One Health approach laid out in the WHO Global Action Plan is appropriate and consistent with statements made in action plans from other international and national organizations. There is however, a long way to go before a fully integrated One Health approach to antimicrobial resistance is implemented at country and global levels. Among the numerous barriers to overcome include the competing interests among multiple sectors (involving animals, humans, and environment) and organizations, agreement on priorities for action, and gaps in antimicrobial resistance surveillance, antimicrobial use policy, and infection control in many parts of the world. 

## 7. Conclusions

History has shown that it is not feasible to neatly separate antimicrobial classes into those exclusively for use in human or non-human sectors, with the exception of new antimicrobial classes. These should probably be reserved for use in humans as long as there are few or no alternatives available. The majority of classes, however, will be available for use in both sectors and the challenge for One Health is to ensure that use of these drugs is optimal overall. This is likely to be achieved when antimicrobials used in both sectors are used for therapy, only rarely for prophylaxis and never for growth promotion, and when we better control the types and amounts of antimicrobials plus the numbers of resistant bacteria we allow to be placed into the environment. What is vitally important is that we do more to stop the spread of resistant bacteria—not only from person to person but between and within the human and agriculture sectors and the environment, giving particular emphasis to controls of contaminated water. 

## Figures and Tables

**Figure 1 tropicalmed-04-00022-f001:**
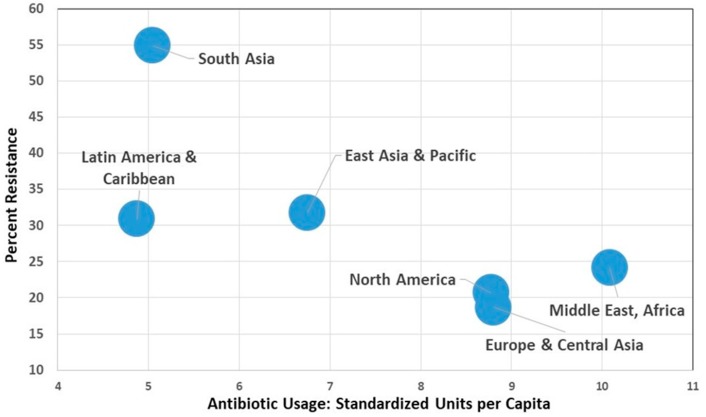
Global aggregated regions: antimicrobial resistance *E. coli* to third generation cephalosporins and fluoroquinolones versus antibiotic usage.

**Figure 2 tropicalmed-04-00022-f002:**
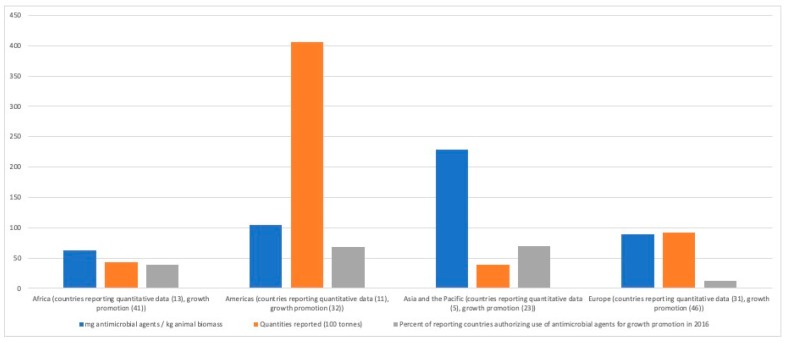
Reported use of antimicrobial agents in animals in 2014 by World Organisation for Animal Health (OIE) Region (adapted from [46]).

**Table 1 tropicalmed-04-00022-t001:** Levels of antibiotic usage in people, resistance levels and other parameters globally. (All data taken from reference 7).

Country	Antibiotic Usage (Standard units per 1000 pop - CCDEP)	*E. coli* % Resistance 3rd gen ceph (WHO)	*E. coli* % Resistance Fluoroquinolones (WHO)	Staphylococcus Aureus (MRSA Rates - WHO)	2015 Corruption Index	GNP per capita 2015 (Purchasing Power Parity in 2011 Dollars)	% with Adequate Sanitation 2015	Improved Water Source (% of Population with Access)
Algeria	15.4	17	2	44.8	36	$13,795	88	87.7
Argentina	6.2	5.1	7.8	54	32	$19,102	96	98.9
Australia	11	7.7	10.6	30	79	$43,631	100	100
Austria	7.2	9.1	22.3	7.4	76	$44,048	100	100
Bahrain		55	62	10	51	$43,754	99	100
Bangladesh	4.3	57.4	89	46	25	$3,137	61	86.2
Belgium	12.6	6	21.5	17.4	77	$41,826	100	100
Bhutan		19.4	35.5	10	65	$7,861	50	100
Bosnia and Herzegovina	7.5	1.5	7.8		38	$10,119	95	99.9
Brazil	5.9	30	40	29.5	38	$14,533	83	98.1
Brunei Darussalam		6.5	12		55	$73,605	100	100
Bulgaria	9.4	22.9	30.2	22.4	41	$17,000	86	99.6
Burkina Faso		36	52.8		38	$1,593	20	82.1
Burundi		7.2	16	13	21	$683	48	75.8
Cambodia		45	71.8		21	$3,278	42	73.4
Canada	7.2	8	26.9	21	83	$42,983	100	99.8
Central African Republic		30	53		24	$581	22	68.4
Chile	4.3	23.8		90	70	$22,197	99	99
China	3	51.9	55.1	38.3	70	$13,572	77	94.8
Colombia	2.9	11.7	59	7.2	37	$12,988	81	91.3
Croatia	10.6	6	14	13	51	$20,664	97	99.6
Cuba		42.9	56		47	$21,017	93	94.6
Cyprus		36.2	47.4	41.6	61	$30,383	100	100
Czech Republic	7.5	11.4	23.5	14.5	56	$30,381	99	100
Denmark	6.7	8.5	14.1	1.2	91	$45,484	100	100
Dominican Republic	2.4	33	49	30	33	$13,372	84	86.5
Ecuador	6.7	15.1	43.8	29	32	$10,777	85	86.9
Egypt	9.1	44.4	34.9	46	36	$10,250	95	99.2
Estonia	4.4		9.9	1.7	70	$27,345	97	99.6
Ethiopia		62	71	31.6	33	$1,530	28	55.4
Finland	7.2	5.1	10.8	2.8	90	$38,994	98	100
France	12.9	8.2	17.9	20.1	70	$37,775	99	100
Germany	7.1	8	23.7	16.2	81	$43,788	99	100
Greece	14.6	14.9	26.6	39.2	46	$24,095	99	100
Guatemala		39.8	41.8	52	28	$7,253	64	92.7
Honduras		36.7	43.1	30	31	$4,785	83	90.6
Hong Kong	7.5				75	$53,463	96	100
Hungary	7.3	15.1	31.2	26.2	51	$24,831	98	100
Iceland		6.2	14		79	$42,704	99	100
India	5	51.4	52.3	42.7	38	$5,733	40	94.1
Indonesia	3.6				36	$10,385	61	86.8
Iran		41	54	53	27	$16,507	90	96.2
Ireland	11.4	9	22.9	23.7	75	$61,378	91	97.9
Israel		2.6	17.9	46.7	61	$31,971	100	100
Italy	11.5	19.8	40.5	38.2	44	$34,220	100	100
Japan	5.3	16.6	34.3	53	75	$37,872	100	100
Jordan	6.3	22.5	14.5		53	$10,240	99	96.9
Kazakhstan	7.5				28	$23,522	98	93.5
Kenya		87.2	91.4	20	25	$2,901	30	63.1
Kuwait	6.3	20.1		32	49	$70,107	100	99
Latvia	5.2	15.9	16.8	9.9	55	$23,080	88	99.3
Lebanon	9.3	27.7	47	20	28	$13,089	81	99
Lesotho		2	14		44	$2,770	30	81.6
Lithuania	7.6	7	12.9	5.8	61	$26,971	92	96.6
Luxembourg	11	8.2	24.1	20.5	81	$93,900	98	100
Malaysia	4.3	17.4	23	17.3	50	$25,312	96	98.2
Malta		12.8	32	49.2	56	$32,720	100	100
Mexico	2.4	42.1	46.3	29.9	35	$16,490	85	96.1
Mongolia		64.1	64.7		39	$11,478	60	64.2
Morocco	6	4	23.3	19	36	$7,365	77	85.3
Myanmar		68	55	26	22	$4,931	80	80.5
Nepal		37.9	64.3	44.9	27	$2,312	46	90.7
Netherlands	4.1	5.7	14.3	1.4	87	$46,354	98	100
New Zealand	10.9	3	6.5	10.4	88	$35,159	100	100
Nicaragua		48.1	42.9		27	$4,884	68	86.9
Nigeria		6.7	36.5	47.1	26	$5,639	29	67.6
Norway	5.9	3.6	9	0.3	87	$63,650	98	100
Pakistan	7.1	36.2	35.3	37.6	30	$4,706	64	91.3
Panama		9.2	23.3	21.1	39	$20,885	75	94.4
Papua New Guinea		24.1	13.3	43.9	25	$2,723	19	40
Paraguay		1.4	22.1	27	27	$8,639	89	96.6
Peru	3.4	44.1	62.8	65.9	36	$11,768	76	86.3
Philippines	2.2	26.7	40.9	54.9	35	$6,938	74	91.5
Poland	9.3	11.7	27.3	24.3	62	$25,323	97	98.3
Portugal	9.3	11.3	27.2	54.6	63	$26,549	100	100
Puerto Rico	9.1						99	
Republic of Moldova		28	15.3	50.3	33	$4,742	76	88.4
Russian Federation	6.2	18	25.7	29.3	29	$24,124	72	96.9
Rwanda		21.4			54	$1,655	62	75.5
Serbia	10.6	21.3	16	44.5	40	$13,278	96	99.3
Singapore	5.7	20	37.8		85	$80,192	100	100
Slovakia	9.2	31	41.9	25.9	51	$28,254	99	100
Slovenia	6.3	8.8	20.7	7.1	60	$29,097	99	99.6
South Africa	8.7	8.2	16.1	52	44	$12,393	66	92.8
South Korea	10.9	24.4	40.9	65.3	56	$34,387	100	97.6
Spain	14.3	12	34.5	22.5	55	$32,219	100	100
Sri Lanka	3.9	58.9	58.8		37	$11,048	95	95.6
Sudan		49.5	56.8		12	$4,121	24	58.5
Saudi Arabia	11.1	15.9	40.9	41.9	52	$50,284	100	97
Sweden	4.8	3	7.9	0.8	89	$45,488	99	100
Switzerland	5.2	8.2	20.2	10.2	86	$56,517	100	100
Syrian Arab Republic		49.8			18	$-	96	90.1
Taiwan	8.7				62			
Thailand	7	37.9	52.5	22.4	38	$15,347	93	97.8
Tunisia	18	20.6	9.4	55.8	38	$10,770	92	97.7
Turkey	18.5	43.3	46.3	31.5	42	$19,460	95	100
United Arab Emirates	10.5	23	32.5	33.4	70	$65,717	98	99.7
United Kingdom	9	9.6	17.5	13.6	81	$38,509	99	100
United States of America	10.3	14.6	33.3	51.3	76	$52,704	100	99.2
Uruguay	6.6	0	15	40	74	$19,952	96	99.6
Venezuela	8.1	12.5	37.2	31	17	$16,769	94	93.1
Vietnam	9.4		0.2		31	$5,667	78	96.4
Zambia		37.4	50.5	32	38	$3,602	44	64.6

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
