# Peer review of "One Health—Its Importance in Helping to Better Control Antimicrobial Resistance"

_tropicalmed, 2019, doi:10.3390/tropicalmed4010022_

Round 1
Reviewer 1 Report
The manuscript entitled "One Health – its importance in helping to better
control antimicrobial resistance" is an intersting review article abput AMR problem. It gives a detailed overview about antibiotic consumption in human and veterinary fields.
My suggestion is to add a graph or table about the current data regarding antibiotic consumption in different regions of the World. You could compare human and veterinary field with a table containing data about antibiotic consumption.
Author Response
Reviewer 1
Comments and Suggestions for Authors
The manuscript entitled "One Health – its importance in helping to better
control antimicrobial resistance" is an interesting review article about AMR problem. It gives a detailed overview about antibiotic consumption in human and veterinary fields.
My suggestion is to add a graph or table about the current data regarding antibiotic consumption in different regions of the World. You could compare human and veterinary field with a table containing data about antibiotic consumption.
Response;
We have now added two figures/graphs and a table (re human use) about the current data regarding antibiotic consumption in different regions of the World involving both human and veterinary fields and using what data is available. The table on human use also has other important parameters including resistance rates and is very large as it contains details on over 100 countries and so has over 100 rows. This could be instead added as a supplement depending on what the editor wants.
Figure 1 has also been submitted in another invited paper (on social factors associated with antibiotic resistance to the journal Antibiotics and which is produced by the same publisher as this journal). The authors for that other paper are Collignon and Beggs. John Beggs has given his permission for us to use the figure he produced in this paper as well. As the same publisher is involved we presume author permission and publisher copyright permission is not needed.
Reviewer 2 Report
This is an interesting paper which further highlights issues with AMR; it helps to collate relevant material in once place, but clarification of the following would be helpful:
line 35 : Darwinian selection does not per se lead do the acquisition or expression of resistant genes - Darwinian selection is the process that drives acquisition or expression
2. line 37: "
The other, and likely more important factor, driving the deteriorating
resistance phenomenon are factors that promote the spread of resistant bacteria" This sentence does not make sense.
3. line 44: "
Wherever antimicrobials are used, there are often already large reservoirs of “resistance” (i.e.
resistant bacteria and resistance genes)." This would be better rendered as "
Wherever antimicrobials are used, there are often already large reservoirs of
resistant bacteria and resistance genes" Why complicate things by adding yet another new term?
4. line 57 Please provide a definition/explanation for One Health
5. Line 128 "
Overall, 101 tonnes of 3rd generation
cephalosporins were used in people Europe (27) and in the US, approximately 82 tonnes in 2011
130 (49)." What year does this apply to for Europe?
Overall, I am left wondering what this article sets out to achieve. It would be helped by an early definition of One Health, along with a clear conclusion about next steps to achieve an integrated One Health policy. The section on WHO One Health policy is a statement of policy, and while it might be laudable, what are the authors trying to say? Do they support it; should it be modified.......
Author Response
Reviewer 2
This is an interesting paper which further highlights issues with AMR; it helps to collate relevant material in once place, but clarification of the following would be helpful:
line 35 : Darwinian selection does not per se lead do the acquisition or expression of resistant genes - Darwinian selection is the process that drives acquisition or expression
Response;
We have deleted this comment as it has the potential to cause confusion for some readers.
2. line 37: "
The other, and likely more important factor, driving the deteriorating
resistance phenomenon are factors that promote the spread of resistant bacteria" This sentence does not make sense.
Response;
We have rephrased this sentence to now read
“Also important in driving the deteriorating resistance problem are factors that promote the spread of resistant bacteria (or “Contagion”)”.
3. line 44: "
Wherever antimicrobials are used, there are often already large reservoirs of “resistance” (i.e.
resistant bacteria and resistance genes)." This would be better rendered as "
Wherever antimicrobials are used, there are often already large reservoirs of resistant bacteria and resistance genes" Why complicate things by adding yet another new term?
Response;
We have reworded this sentence to read
“Wherever antimicrobials are used, there are often already large reservoirs of resistant bacteria and resistance genes.”
4. line 57 Please provide a definition/explanation for One Health
Response;
We have added –
The One Health is defined by WHO (ref) and others (refs) as a concept and approach to “designing and implementing programmes, policies, legislation and research in which multiple sectors communicate and work together to achieve better public health outcomes. The areas of work in which a One Health approach is particularly relevant include food safety, the control of zoonoses and combatting antibiotic resistance” (ref). It needs to involve the “collaborative effort of multiple health science professions, together with their related disciplines and institutions – working locally, nationally, and globally – to attain optimal health for people, domestic animals, wildlife, plants, and our environment” (ref).
Main refs will be WHO and One Health Commission. 2018. https://www.onehealthcommission.org/en/why_one_health/what_is_one_health/
5. Line 128 "
Overall, 101 tonnes of 3rd generation
cephalosporins were used in people Europe (27) and in the US, approximately 82 tonnes in 2011
130 (49)." What year does this apply to for Europe?
Response;
2012 – now added
Overall, I am left wondering what this article sets out to achieve. It would be helped by an early definition of One Health, along with a clear conclusion about next steps to achieve an integrated One Health policy. The section on WHO One Health policy is a statement of policy, and while it might be laudable, what are the authors trying to say? Do they support it; should it be modified.......
Response;
We have added the definition as suggested by the reviewer.
Also we have now added before the conclusion:
“The One Health approach laid out in the WHO Global Action Plan is appropriate and consistent with statements made in action plans from other international and national organizations. There is however, a long way to go before a fully integrated One Health approach to antimicrobial resistance is implemented at country and global levels. Among the numerous barriers to overcome include the competing interests among multiple sectors (involving animals, humans, and environment) and organizations, agreement on priorities for action, and gaps in antimicrobial resistance surveillance, antimicrobial use policy, and infection control in many parts of the world. “